# Phosphatidylserine from *Portunus*
*trituberculatus* Eggs Alleviates Insulin Resistance and Alters the Gut Microbiota in High-Fat-Diet-Fed Mice

**DOI:** 10.3390/md18090483

**Published:** 2020-09-22

**Authors:** Shiwei Hu, Mengyu Du, Laijin Su, Huicheng Yang

**Affiliations:** 1Innovation Application Institute, Zhejiang Ocean University, Zhoushan 316022, China; hushiweihai@163.com (S.H.); dumengyu816@163.com (M.D.); 2Wenzhou Academy of Agricultural Science, Wenzhou Characteristic Food Resources Engineering and Technology Research Center, Wenzhou 325006, China; 3Zhejiang Marine Development Research Institute, Zhoushan 316021, China; huichengy@163.com

**Keywords:** phosphatidylserine, *Portunus trituberculatus* eggs, insulin resistance, gut microbiota

## Abstract

*Portunus trituberculatus* eggs contain phospholipids, whose components and bioactivity are unclear. Here, we investigated the fatty acid composition of phosphatidylserine from *P. trituberculatus* eggs (*Pt*-PS). Moreover, its effects on insulin resistance and gut microbiota were also evaluated in high-fat-diet-fed mice. Our results showed that *Pt*-PS accounted for 26.51% of phospholipids and contained abundant polyunsaturated fatty acids (more than 50% of eicosapentaenoic acid (EPA) and docosahexaenoic acid (DHA)). Animal experiments indicated that *Pt*-PS significantly decreased body weight and adipose weight gain, improved hyperglycemia and hyperinsulinemia, mitigated insulin resistance, and regulated circulatory cytokines. *Pt*-PS activated insulin receptor substrate 1 (IRS1) and increased the levels of IRS1-associated phosphatidylinositol 3-hydroxy kinase (PI3K), phosphorylated protein kinase B (Akt) protein, and plasma membrane glucose transporter 4 protein. Furthermore, *Pt*-PS modified the gut microbiota, inducing, especially, a dramatic decrease in the ratio of Firmicutes to Bacteroidetes at the phylum level, as well as a remarkable improvement in their subordinate categories. *Pt*-PS also reduced fecal lipopolysaccharide concentration and enhanced fecal acetate, propionate, and butyrate concentrations. Additionally, the effects of *Pt*-PS on alleviation of insulin resistance and regulation of intestinal bacteria were better than those of phosphatidylserine from soybean. These results suggest that *Pt*-PS mitigates insulin resistance by altering the gut microbiota. Therefore, *Pt*-PS may be developed as an effective food supplement for the inhibition of insulin resistance and the regulation of human gut health.

## 1. Introduction

Insulin resistance is a serious polygenic disease that concerns approximately 90% of diabetic patients and also underlies the development of several other metabolic disorders, including cardiovascular disease and certain cancers [1,2]. Once the circulating insulin fails to balance body glucose homeostasis, insulin resistance occurs. About 75% of insulin-disposed glucose is observed in skeletal muscle [3]. This procedure is mainly mediated through phosphatidylinositol 3-hydroxy kinase (PI3K)/protein kinase B (Akt) signal transduction [4]. In skeletal muscle, insulin receptor substrate (IRS) is activated by insulin stimulation through inhibition of phosphorylation at its serine site and elevation of phosphorylation at its tyrosine site [5]. Activated IRS polymerizes PI3K and triggers the phosphorylation of the p85 regulatory subunit [6]. The signal is subsequently transmitted downstream, leading to Akt phosphorylation and glucose transporter 4 (Glut4) translocation from the cytoplasm to the cytomembrane [6,7]. Glut4 in the plasma membrane (m-Glut4) is the exclusive protein that transports glucose into cells [8]. In addition, increasing evidence strongly suggests that the etiology or development of insulin resistance is closely associated with the gut microbiota [9,10,11,12]. Particularly, increased amounts of Firmicutes and decreased amounts of Bacteroidetes positively lead to the development of insulin resistance [13]. Structure imbalance in the gut microbiota directly elevates the concentration of lipopolysaccharide (LPS) in circulating systems and reduces the levels of short-chain fatty acids (SCFAs) [14], which induces insulin resistance by triggering the PI3K/Akt insulin pathway or by influencing the levels of circulating adipokines [15,16].

The swimming crab, *Portunus trituberculatus*, has become a significant economic marine product in China because of its high nutritional value and extensive productions. Its production in China amounted to 100,000 tons in 2015 and 600,000 tons in 2017, and the production capacity is steadily growing [17,18]. There are few studies on the processing or utilization of *P. trituberculatus*, while several papers have focused on its gene sequence analysis or aquaculture [19,20]. Recently, we revealed that egg oil isolated from *P. trituberculatus* (*Pt*-egg oil) contains abundant phospholipids and improved hyperglycemia and hyperinsulinemia in high-fat-diet (HFD)-fed mice [21]. Furthermore, *Pt*-egg oil also showed anti-obese effects through regulating the gut microbiota [22]. However, the composition of phospholipids in *P. trituberculatus* is unclear, and the bioactivity of phospholipids in this crab should be uncovered for the exploitation and utilization of this marine product. Here, we isolated phosphatidylserine from *P. trituberculatus* eggs (*Pt*-PS), analyzed its fatty acid composition, and investigated its effects on insulin resistance and the gut microbiota.

## 2. Results

### 2.1. Fatty Acid Composition of Pt-PS

As shown in Table 1, we identified 18 kinds of fatty acids in *Pt*-PS and found a greater proportion of unsaturated fatty acids (UFA) than of saturated fatty acids (69.29 ± 5.20% vs. 22.19 ± 3.15%). Polyunsaturated fatty acids (PUFA) accounted for 82.45% of UFA, whereas monounsaturated fatty acids only made up 17.55%. Importantly, eicosapentaenoic acid (EPA) and docosahexaenoic acid (DHA) appeared to be abundant in the fatty acids of *Pt*-PS, corresponding approximately to 50.13% of them.

### 2.2. Pt-PS Protects Mice from HFD-Induced Obesity

HFD-induced obesity contributes to insulin resistance. Through all the intervention stage, mice fed with HFD showed significant body weight gain compared with animals in the control group (*p* < 0.01), which was accompanied by higher food intake and calorie intake (Table 2). There was a remarkable decrease in body weight gain of *Pt*-PS-treated mice, corresponding to 33.75%, compared with the HFD group (*p* < 0.01), with no significant changes in food and calorie intake. PS from soybean (Sb-PS) also markedly reduced body weight gain (*p* < 0.05), but its effect was inferior to that of *Pt*-PS (20.52 ± 1.86 vs. 16.61 ± 1.44). In addition, *Pt*-PS supplementation dramatically inhibited the development of both adipose tissue and hepatic tissue in the *Pt*-PS group compared with the HFD group (perirenal fat, 51.56%, *p* < 0.01; epididymal fat, 48.68%, *p* < 0.01; abdominal fat, 51.09%, *p* < 0.01; liver 30.26%, *p* < 0.01). Similar to what observed for body weight gain, the effects of *Pt*-PS on the suppression of adipose and hepatic tissues were superior to those of Sb-PS (perirenal fat, 0.31 ± 0.02 vs. 0.41 ± 0.03; epididymal fat, 1.17 ± 0.16 vs. 1.84 ± 0.21; abdominal fat 0.45 ± 0.08 vs. 0.65 ± 0.07; liver, 1.36 ± 0.15 vs. 1.50 ± 0.20).

### 2.3. Pt-PS Improves Insulin Resistance

Mice in the HFD group exhibited higher levels of fasting blood glucose and serum insulin, as well as higher values of insulin resistance index (HOMA-IR) and quantitative insulin sensitivity check index (QUICKI) than those in the control group (*p* < 0.01). Fasting blood glucose, serum insulin, and HOMA-IR score were significantly decreased in obese mice when treated with *Pt*-PS by 29.39%, 25.78%, and 47.83%, respectively (*p* < 0.01), while the QUICKI value increased by 13.33% (*p* < 0.05). Sb-PS also caused significant increases in fasting blood glucose, serum insulin, and HOMA-IR score by 22.56% (*p* < 0.05), 19.09% (*p* < 0.05), and 39.00% (*p* < 0.01), respectively, but no remarkable changes in QUICKI value. Additionally, impaired glucose tolerance and insulin tolerance by HFD were both restored when obese mice were treated with *Pt*-PS and Sb-PS (*p* < 0.05 and *p* < 0.01, Figure 1). The oral glucose tolerance test (OGTT) and intraperitoneal insulin tolerance test (IITT) showed that the decreases of AUC_OGTT_ and AUC_IITT_ in *Pt*-PS-treated mice were greater than in Sb-PS-fed animals (AUC_OGTT_, 26.77 ± 2.48 vs. 30.37± 2.82; AUC_IITT_, 14.35 ± 1.43 vs. 16.15± 1.48). This distinctly suggested that the improvement in insulin resistance by *Pt*-PS was clearly greater than that induced by Sb-PS.

### 2.4. Pt-PS Regulates Serum Adipokines

Adipokines affect insulin resistance both by inducing insulin secretion directly and by regulating PI3K/Akt signaling mediated by insulin indirectly. As shown in Table 2, serum adiponectin was significantly increased by 96.03% in the *Pt*-PS group compared with the HFD group (*p* < 0.01) and by 75.11% in the Sb-PS group (*p* < 0.01). Serum tumor necrosis factor-α (TNF-α), leptin, and resistin were significantly reduced by 31.73% (*p* < 0.05), 47.83% (*p* < 0.01), and 47.25% (*p* < 0.01), respectively, in *Pt*-PS-supplemented mice compared with insulin resistant animals. Sb-PS also caused a marked decrease in serum resistin, corresponding to 30.28% (*p* < 0.01), but no significant changes in serum TNF-α and leptin. This suggested that the effects of *Pt*-PS on serum adipokines were greater than those of Sb-PS.

### 2.5. Pt-PS Activates the PI3K/Akt Pathway in Skeletal Muscle

The classical PI3K/Akt insulin signal pathway in skeletal muscle was investigated by western blotting. As shown in Figure 2, mice in the *Pt*-PS group showed a dramatic decrease in phosphorylated IRS1 protein at the serine site and an increase in phosphorylated IRS1 protein at the tyrosine site (*p* < 0.01). In contrast, Sb-PS only induced a significant decrease in p-IRS1(ser) protein (*p* < 0.05), with no significant change in p-IRS1(tyr) protein. Meanwhile, both *Pt*-PS and Sb-PS increased PI3K–IRS1 protein by 6.77-fold and 5.11-fold, respectively (*p* < 0.01). Tyrosine phosphorylated Akt protein was obviously increased in the *Pt*-PS group compared with the HFD group (*p* < 0.01), while no marked alteration was observed in the Sb-PS group. Interestingly, both *Pt*-PS- and Sb-PS-supplemented mice exhibited a notable increase in m-Glut4 protein by 3.32 fold and 1.19 fold, respectively (*p* < 0.01).

### 2.6. Pt-PS Restores Gut Microbiota Dysbiosis

Gut microbiota dysbiosis stimulates the development and exacerbation of insulin resistance. Here, we investigated the effects of *Pt*-PS on the maintenance of the gut bacteria community in mice with HFD-induced insulin resistance through analysis of the V3–V4 fragment of the 16S rRNA gene from fecal samples. The Chao index is usually considered to reflect within-community richness. As shown in Figure 3a, the Chao index of operational taxonomic units (OTU) in the HFD group was significantly lower compared with that of normal animals (*p* < 0.001). *Pt*-PS treatment caused a significant increase in Chao index with respect to that of insulin-resistant mice (*p* < 0.05), while there was no significant difference in Chao index between the HFD group and the Sb-PS group. This suggested that *Pt*-PS treatment recovered the richness of the bacteria species, which was reduced in insulin-resistant mice. As regards alpha diversity, Shannon and Simpson indexes are generally used to express community diversity. As shown in Figure 3b,c, there was a remarkable increase in the Shannon index and an obvious decrease in the Simpson index in the *Pt*-PS group compared with the HFD group (*p* < 0.05), but Sb-PS had no significant effect on the two indexes. This implies that the diversity of intestinal bacteria was prominently increased by *Pt*-PS. The community structure and relationships in the four groups were assessed using Principal Coordinates Analysis (PCoA) at the OTU level (Figure 3e). The gut microbiota was distinct in each group, except for three animals of the Sb-PS group whose gut microbiota tended to overlap with that of the HFD group. Hierarchical clustering at the OTU level also showed distinct gut microbial compositions (Figure 3d). Collectively, this indicated that *Pt*-PS dramatically modulated the gut microbiota in insulin-resistant mice.

The changes of microbial composition induced by *Pt*-PS treatment were further assessed at various taxonomic levels. Data of microbial composition at the phylum level are shown in Figure 4. There was a significant increase in Firmicutes and an obvious decrease in Bacteroidetes. The *Pt*-PS supplement was found to considerably modify the relative abundances of Firmicutes and Bacteroidetes (*p* < 0.01). Interestingly, *Pt*-PS dramatically reduced the ratio of Firmicutes to Bacteroidetes (*p* < 0.01). Moreover, the ratio of Firmicutes to Bacteroidetes was also significantly lowered in the Sb-PS group compared with the HFD group (*p* < 0.05), though no significant changes in the abundances of Firmicutes and Bacteroidetes were observed.

Figure 5 shows the community ternary phase diagram and the relative microbial abundances at the order level. Compared with the HFD group, *Pt*-PS markedly enhanced the abundances of Bacteroidetes, Pseudomonadales, and Verrucomicrobiales (*p* < 0.01), while it reduced the abundances of Lactobacillales, Erysipelotrichales, Coriobacteriales, and Desulfovibrionales (*p* < 0.01). The abundances of Clostridiales and Bifidobacteriales between HFD and *Pt*-PS groups did not show changes. Sb-PS treatment also caused a significant increase in Pseudomonadales (*p* < 0.01) and a marked decrease in Erysipelotrichales (*p* < 0.05) compared with the HFD group.

Furthermore, community heatmap analysis at the genus level was also used to determine the relative microbial abundances of the gut microbiota (Figure 6). All 40 main bacterial genera showed remarkable differences when comparing insulin-resistant mice and control animals, while 39 genera differed between *Pt*-PS and HFD groups, suggesting that the protective activity of *Pt*-PS on insulin resistance was regulated by bacterial subsets of different genera. *Pt*-PS treatment decreased the abundance of *Helicobacter* (belonging to Proteobacteria), *Ruminococcaceae UCG_014*, *Faecalibaculum*, *norank_f_Ruminococcaceae*, *[Ruminococcus]_torques_group*, *norank_f_Erysipelotricheceae* (all belonging to Firmicutes), *Mucispirillum*, *[Eubacterium]_xylanophilum_group*, *Allobaculum*, *Lactobacillum*, *Clostridium_sensu_stricto_1*, *Romboutsia*, *Parvibacter*, *Cloriobacteriaceae_UCG_002*, *Desulfovibrio*, *Enterorhabdus*, and *norank_f_Lachnospiraceae* in insulin-resistant mice. Moreover, *Pt*-PS increased the number of *Odoribacter*, *Escherichia_shigella*, *Family_ XIII_AD3011_group*, *Streptococcus*, *Parabacteroides*, *Paraprevotella*, *Lachnoclostridium*, *Enterobacter*, *[Eubacterium]_fissicatena_group*, *Drysipelatoclostridium*, *Alistripes*, and *Pselldomonas*. Interestingly, the number of the SCFAs-producing bacteria *Ruminococcaceae_UCG_013*, *Unclassified_f_Ruminococcaceae*, *Bacteroides*, *norank_f_Bacteroidales_S24-7_group*, *[Eubacterium]_coprostanoligenes_group*, *Unclassified_f_Lachnospiraceae*, *Faecalitalea*, *Bifidobacterium*, *Akkermansia*, and *Alloprevotella* was increased in HFD-fed mice when supplemented with *Pt*-PS.

### 2.7. Pt-PS Decreases LPS Concentration and Increases SCFAs Concentrations

As shown in Table 3, HFD caused a significant increase in fecal LPS level as compared with control mice (*p* < 0.01), while supplementation of *Pt*-PS significantly lowered HFD-induced LPS appearance in the feces by 44.10% (*p* < 0.01). Moreover, fecal acetate, propionate, and butyrate concentrations were remarkably elevated by 59.66% (*p* < 0.01), 43.12% (*p* < 0.05), and 1.13 fold (*p* < 0.01), respectively, in the *Pt*-PS group compared with insulin-resistant mice. There was also a 29.09% increase in fecal acetate level in mice treated with Sb-PS (*p* < 0.05), while no significant changes in fecal LPS, propionate, and butyrate concentrations were observed.

## 3. Discussion

The components and bioactivity of phospholipids in *P. trituberculatus* eggs are poorly known. Here, we focused on *Pt*-PS, whose content in phospholipids amounts to 26.51%. PUFA were found to be abundant in the fatty acids of *Pt*-PS, especially EPA and DHA. In mice, *Pt*-PS significantly reduced body weight gain, adipose and hepatic tissues weight, fasting blood glucose, serum insulin, and HOMA-IR score, increased QUICKI value, and regulated insulin resistance-related adipokines. This suggests that *Pt*-PS can alleviate insulin resistance and enhance insulin sensitivity.

Marine animals contain biologically interesting phospholipids. Phosphatidylcholine (PC) is the predominant lipid in most marine animals [23]. It was reported that PC content corresponds to about 80% of phospholipids in salmon roe and to 76% in large yellow croaker roe [24,25], but only to 11–15% in sea bass eggs (whose predominant phospholipid is phosphatidylinositol, corresponding approximately to 47–66%) [26]. In *P. trituberculatus* eggs’ phospholipids, PC is the dominant phospholipid, amounting to 32.28% of them (data not shown). The second abundant phospholipid is PS (26.51%); this is interesting because PS is present in traces or is inexistent in other marine animals [23]. Several studies have proved that PS has superior bioactivity than PC, due to the presence of a serine head group instead of a choline head group [27,28,29,30]. Therefore, we investigated the effects of *Pt*-PS.

A typical feature of phospholipids from marine foods is that they contain abundant PUFA, particularly EPA and DHA. For example, researchers found 21.01% of EPA + DHA in PE and 36.00% of EPA + DHA in PC from the sea cucumber, *Apostichopus japonicus* [31]. EPA + DHA account for 41.90% of the phospholipids from the eggs of the large yellow croaker, *Pseudosciaena crocea* [25]. Our data showed over 50% of EPA + DHA in the fatty acids of *Pt*-PS. The phospholipids from soybean are mainly rich in linoleic acid. In this study, we compared the effects of *Pt*-PS and Sb-PS, and the results showed that the beneficial activity of *Pt*-PS on insulin resistance and on the regulation of the gut microbiota was much greater than that of Sb-PS. This indicates that EPA and DHA bonding in PS is crucial for the activity of *Pt*-PS.

Glucose entrance into skeletal muscle cells is the foremost process of glucose disposal. In this process, Glut4 protein translocation from intracellular sites to the cell surface is pivotal. It is controversial whether this process depends on the total glucose transporter protein. Gandhi et al. reported that it is achieved by Glut4 protein translocation, rather than total protein [32], and Kim et al. also reported that total Glut4 protein was unchanged in control and insulin-resistant mice [33]. However, other papers showed that an increase in total Glut4 protein positively contributed to the entrance of glucose into cells and the improvement of insulin resistance [34,35]. Here, we detected the expression of m-Glut4 protein. A HFD could directly induce a significant decrease in m-Glut4 protein expression, while *Pt*-PS supplementation significantly elevated the expression of key proteins in the plasma membrane of skeletal muscle cells. Glut4 translocation is predominantly triggered by PI3K/Akt signaling [36], in which several proteins are involved, including phosphorylated IRS1, PI3K–IRS1 and phosphorylated Akt. Hatem-Vaquero et al. reported that obvious peripheral insulin resistance resulted from the inactivation of the PI3K/Akt pathway and the reduction of Glut4 expression in ILK-depleted mice [37]. Inversely, the activation of the PI3K/Akt pathway could promote Glut4 translocation and attenuated hepatic insulin resistance in type 2 diabetic rats [38]. In the present study, when treated with *Pt*-PS, HFD-fed mice showed an improvement of insulin resistance and the activation of the PI3K/Akt pathway, suggesting that *Pt*-PS mitigated insulin resistance through the activation of PI3K/Akt signaling in skeletal muscle. In addition, we also detected adipokines, which play an important role in the development of insulin resistance. The anti-inflammatory adipokine adiponectin was increased in *Pt*-PS-treated mice. Adiponectin exerts an insulin-sensitizing activity by binding to its receptors and finally activates AMPK, PPAR-α, and IRS [39]. Leptin, acting through LRb, has been shown to regulate IRS, PI3K, and Akt, raising the possibility of an interaction between leptin and insulin [40]. TNF-α could directly stimulate the phosphorylation of IRS at serine residues, which inactivates the downstream PI3K/Akt signaling [41]. The secretion of resistin can be upregulated by insulin and glucose, and the exposure of rodent to resistin results in a decreased response to insulin, which finally interferes with the activation of IRS and PI3K [42]. In this study, serum resistin, leptin, and TNF-α levels were dramatically decreased by *Pt*-PS supplementation, which was associated with the alleviation of insulin resistance. These findings indicate that *Pt*-PS alleviated insulin resistance, at least in part, via normalization of circulatory adipokines.

Gut microbiota dysbiosis is closely related to insulin resistance and they affect each other [43]. The increase in the ratio of Firmicutes to Bacteroidetes in the gut microbiota is considered an important sign of insulin resistance [44]. In the present study, *Pt*-PS prevented HFD-dependent alteration of intestinal bacteria at the phylum level, increasing Bacteroidetes and decreasing Firmicutes and the ratio of Firmicutes to Bacteroidetes. Moreover, *Pt*-PS caused a marked reduction of destructive gut bacteria belonging to Firmicutes, such as *Ruminococcaceae UCG_014*, *Faecalibaculum*, *norank_f_Ruminococcaceae*, *[Ruminococcus]_torques_group*, and *norank_f_Erysipelotricheceae*. *Bacteroides* and *norank_f_Bacteroidales_S24-7_group*, belonging to Bacteroidetes, were elevated in the feces of *Pt*-PS-treated mice. These changes of the gut microbiota were associated with the improvement of insulin resistance, indicated by a normalization of blood and insulin parameters. Our results are strongly supported by numerous papers. Yue et al. reported that theabrownin could significantly reduce the Firmicutes/Bacteroidetes ratio and improve insulin resistance in rats with a high-sugar diet [45]. Monk et al. proved that fish oil supplementation remarkably normalized serum adipokines concentrations and insulin resistance by decreasing the abundance of Firmicutes and increasing the levels of Bacteroidetes members [46]. Liu et al. showed that oral hydroxysafflor yellow A significantly ameliorated insulin resistance and increased SCFAs production by modulating the gut microbiota in HFD-fed mice, which included a decrease in the Firmicutes/Bacteroidetes ratio and an increase in the amount of several SCFAs-producing bacteria [47]. This indicates that *Pt*-PS-altered gut microbiota is a pivotal factor to alleviate insulin resistance.

Gut microbiota can be associated with insulin resistance by metabolites, including LPS and SCFAs. The imbalance of microbial composition directly increases endotoxin levels (such as LPS) and lowers the concentrations of SCFAs, which provokes excessive levels of adipokines, insulin resistance, and even diabetes [48,49]. In this study, compared with insulin-resistant mice, *Pt*-PS-treated animals showed a dramatic reduction of fecal LPS concentration, which may be associated with a decrease in the number of LPS producers (Gram-negative bacteria), such as *Helicobacter* and *Desulfovibrio* [50]. On the other hand, acetate, propionate, and butyrate were generated from the fermentation of dietary fiber by *Ruminococcaceae*, *Lachnospiraceae*, *Akkermansia*, *Bifidobacterium*, etc. [51,52]. In the present study, *Pt*-PS supplementation significantly increased the three aforementioned SCFAs, which was associated with increases in the abundance of *Ruminococcaceae_UCG_013*, *Unclassified_f_Ruminococcaceae*, *Bacteroides*, *Faecalitalea*, *norank_f_Bacteroidales_S24-7_group*, *[Eubacterium]_coprostanoligenes_group*, *Unclassified_f_Lachnospiraceae*, *Bifidobacterium*, *Akkermansia*, and *Alloprevotella*. This demonstrates that *Pt*-PS-induced SCFAs generation through modulation of the gut microbiota improves insulin resistance in mice.

## 4. Materials and Methods

### 4.1. Preparation of Pt-PS

The lipids were extracted from *Pt* eggs according our previous paper [22] and subsequently used to isolate PS. Briefly, total lipids were extracted from *P. trituberculatus* eggs using chloroform/methanol and subsequently isolated using silica column chromatography to obtain phospholipids. Using high-performance liquid chromatography (HPLC), the composition of phospholipids was determined and resulted to include phosphatidylcholine, PS, phosphatidic acid, phosphatidylinositol, phosphatidylethanolamine, and others, PS content was 26.51%. *Pt*-PS was isolated from the phospholipids using HPLC (1200, Agilent, Santa Clara, CA, USA). The fatty acid composition of *Pt*-PS was determined using gas chromatography (7820A, Agilent, Santa Clara, CA, USA).

### 4.2. Animal Experiments

Male C57BL/6J mice, 8 weeks of age, were purchased from Vital River Laboratory Animal Center (SCXK2016-0001, Beijing, China). All experimental mice were housed in individual cages. All procedures were reviewed and approved by the Ethics Committee at the Zhejiang Ocean University. The animals were divided into four groups (*n* = 14): Control group (fed with standard diet), HFD group (fed with a HFD: 29% carbohydrates, 16% protein, and 55% fat), Sb-PS group (maintained with HFD and Sb-PS at a dosage of 1%), and *Pt*-PS group (administrated HFD and *Pt*-PS at a dosage of 1%). After 16 weeks of continuous treatment, each animal was fed in separated metabolic cages for 24 h to collect feces and then was sacrificed to collect blood and skeletal muscle for further study. Figure 7 shows the animal experimental procedure.

### 4.3. OGTT and IITT

At 15 weeks of administration, the OGTT was carried out by quantifying the blood glucose levels of 6 h fasted mice at 0, 0.5, 1, and 2 h after intragastric administration of 2 g/kg glucose (*n* = 7). IITT was performed similarly with an intraperitoneal injection of 0.5 U/kg insulin using the remaining animals. Blood glucose levels were measured using a commercial kit. The areas under curve of OGTT and IITT (AUC_OGTT/IITT_) were both calculated using Equation (1):AUC_OGTT_/AUC_IITT_ = 0.25 × A + 0.5 × B +0.75 × C + 0.5 × D(1)

(A, B, C, and D represent blood glucose level at 0, 0.5, 1, and 2 h after treating with glucose or insulin, respectively)

### 4.4. Insulin Stimulation and Plasma Membrane Preparation

Insulin stimulation was performed to analyze the phosphorylated proteins in PI3K/Akt signaling pathway and plasma membrane Glut4 protein, according to our previous paper [21]. Briefly, for phosphorylated proteins, 6 h fasted mice were intraperitoneally injected with 40 U/kg insulin and then sacrificed to strip skeletal muscle 5 min after the injection (*n* = 4). For plasma membrane Glut4, a similar treatment was carried out using a 0.5 U/kg insulin injection and waiting 30 min (*n* = 3). The other mice were injected with normal saline (*n* = 7). The skeletal muscle was frozen in liquid nitrogen and stored at −80 °C for subsequent study.

The skeletal muscle (3 g) was stripped from 0.5 U/kg insulin-injected mice and subsequently homogenized in normal saline. Crude membrane was obtained with triple centrifugation at 1200× *g*, 9000× *g*, and 19,000× *g*. The crude membrane was then centrifuged using 25%, 32%, and 35% sucrose-density-gradient centrifugation to separate plasma membrane fractions at 150,000× *g* for 16 h, and the fraction of 25% sucrose solution was centrifuged at 190,000× *g* for 1 h to collect plasma membrane Glut4 (m-Glut4) protein.

### 4.5. Analysis of Insulin Resistance-Related Parameters

Blood was collected from the mice injected with normal saline to measure the fasting blood glucose level. Serum was obtained from the blood by centrifugation at 7500× *g* for 15 min to determine the insulin level. HOMA-IR and QUICKI were calculated according to Equations (2) and (3):HOMA-IR = (fasting blood glucose × serum insulin)/22.5(2)
QUICKI = 1/[1 g(fasting blood glucose) + 1 g(serum insulin)](3)

### 4.6. Adipokines Measurement

Four insulin resistance-related adipokines in serum were detected using the corresponding ELISA kits (Invitrogen, Carlsbad, CA, USA), namely, adiponectin, resistin, leptin, and TNF-α.

### 4.7. Immunoprecipitation and Western Blotting

IRS1-associated PI3K (PI3K–IRS1) was isolated by immunoprecipitation, according to our previous study [21]. Briefly, the homogenization of skeletal muscle treated with normal saline was prepared in phosphate-buffered saline. Total proteins were obtained from the cell lysates using centrifugation, then were incubated with an anti-IRS1 antibody, and subsequently were precipitated using protein A-sepharose. The immunocomplexes were used to analyze PI3K–IRS1 through western blotting.

The skeletal muscle (0.1 g) was lysed in IP lysis buffer to dissolve cellular proteins. Proteins, including cellular proteins, PI3K–IRS1 protein, and m-Glut4 protein, were separated by sodium dodecyl sulfate-polyacrylamide gel electrophoresis, transferred to polyvinylidene fluoride membranes, incubated with primary antibodies and subsequently HRP-IGG secondary antibodies, and developed with chemiluminescent autography. Phosphorylated proteins and m-Glut4 protein were evaluated by comparison to their corresponding proteins, while PI3–-IRS1 protein was normalized by IRS1 protein.

### 4.8. Fecal LPS and SCFAs Determination

Feces from the normal saline-injected mice were homogenized in ice-cold H_2_O. The supernatant was heated to 70 °C to inactivate proteins, and LPS concentration was evaluated by an ELISA kit.

Fecal SCFAs, including acetate, propionate, and butyrate, were detected according to our previous paper. Briefly, the homogenization of feces from normal saline-treated mice was performed with 12% formic acid (contained 1 mM 2-ethylbutyric acid, pH 2.5). The supernatant was filtered using 0.22 µm polytetrafluoroethylene syringe filters, and SCFAs were tested through GC/MS (5975-7890A Agilent, Santa Clara, CA, USA).

### 4.9. Fecal DNA Extraction and Microbiota Analysis

Genomic DNA of feces from normal saline-injected mice was extracted by QIAamp DNA Stool Mini Kit (Qiagen, Dusseldorf, Germany), and its concentration and purity were expressed by absorbance at 260 nm and the ratio of A_260_ to A_280_, respectively.

The extracted DNA was used as a template to amplify the V3 + V4 hyper-variable region of bacterial 16S rRNA with the universal primers 341F (5′-CCTAYGGGRBGCASCAG-3′) and 806R (5′-GGACTACNNGGGTATCTTAAT-3′). After purifying and sequencing using an Illumina HiSeq platform, high-quality paired-end reads with average quality > Q20 were used for subsequent study. The tags were aligned into OTU according to a 97% similarity, and the bacterial diversity was analyzed through OTU table and phylogenetic tree.

### 4.10. Statistical Analysis

Data are expressed as the mean ± standard deviation (SD). One-way analysis of variance (ANOVA) followed by Student’s *t*-test was performed to determine significant differences among the four groups with SPSS 21.0 software. Statistical significance was considered at *p* < 0.05.

For bacterial sequence data, Shannon index, Simpson index, and Chao index, which indicate alpha diversity, were analyzed using Student’s *t*-test. Univariate differential abundance of OTU at the phylum, order, family, and genus levels was verified by incorporating Fisher’s test. *p* values were generated through Metastats and corrected using the Benjamini–Hochberg method. Statistical significance was set at *p* < 0.05.

## 5. Conclusions

*Pt*-PS was isolated from *P. trituberculatus* eggs and resulted to contain abundant PUFA, in particular EPA and DHA. The present study also indicates that *Pt*-PS significantly mitigates insulin resistance in HFD-fed mice. Normalization of circulatory adipokines and activation of the PI3K/Akt insulin signal cascade in skeletal muscle was found to be the underlying mechanism for *Pt*-PS-alleviated insulin resistance. Our study also demonstrated interactions between dietary *Pt*-PS and the gut microbiome as a novel mechanism underlying the anti-insulin resistance activity of *Pt*-PS. Our study has important implications for the utilization of phospholipids from swimming crab as an alternative marine drug to improve insulin resistance.

## Figures and Tables

**Figure 1 marinedrugs-18-00483-f001:**
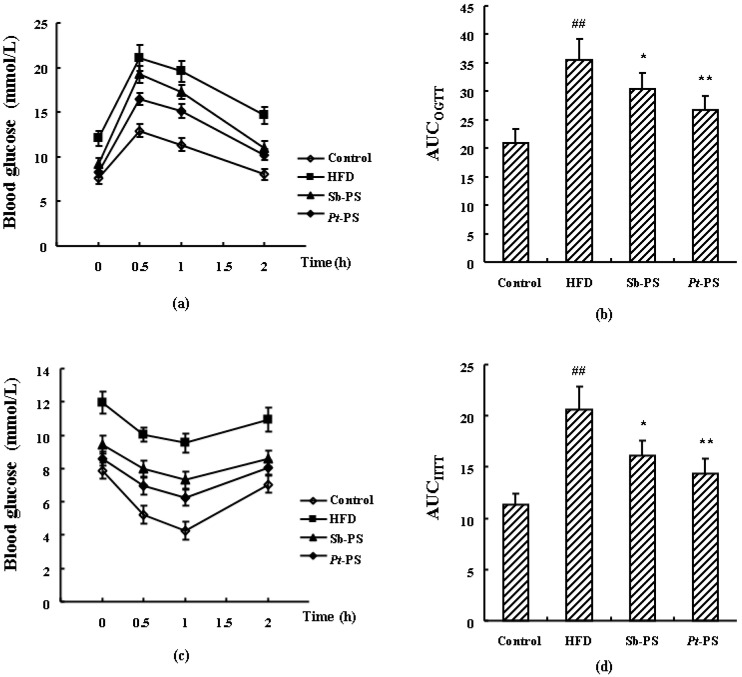
Effects of *Pt*-PS determined in the oral glucose tolerance test (OGTT) and intraperitoneal insulin tolerance test (IITT) in HFD-fed mice (*n* = 7). (**a**) Blood glucose level at indicated time points measured in the OGTT; (**b**) AUC_OGTT_; (**c**) blood glucose level at indicated time points measured in the IITT; (**d**) AUC_IITT_. Data are expressed as mean ± SD. ^##^
*p* < 0.01 vs. control mice; * *p* < 0.05, ** *p* < 0.01 vs. HFD mice.

**Figure 2 marinedrugs-18-00483-f002:**
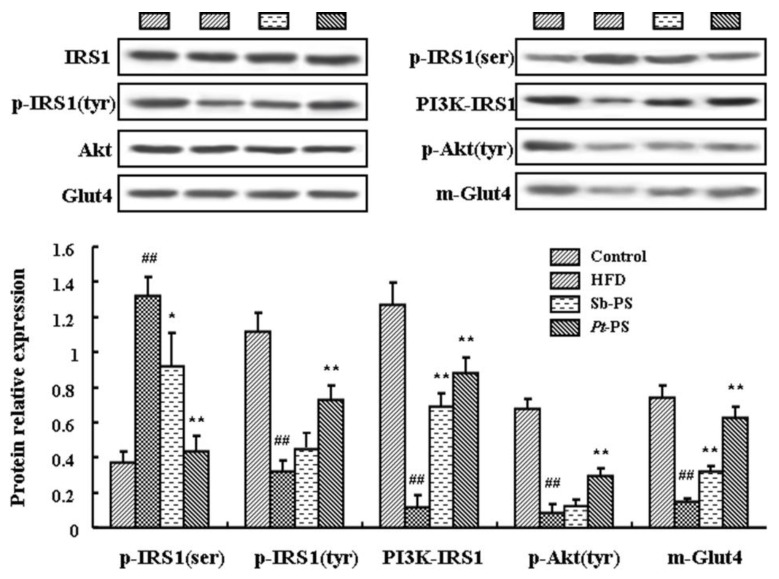
Effects of *Pt*-PS on the expression of key proteins in the PI3K/Akt pathway. Levels of phosphorylated proteins and PI3K–IRS1 protein (*n* = 4), m-Glut4 protein (*n* = 3), IRS1, Akt, and Glut4 protein (*n* = 7) were determined. ^##^
*p* < 0.01 vs. control; * *p* < 0.05, ** *p* < 0.01 vs. HFD.

**Figure 3 marinedrugs-18-00483-f003:**
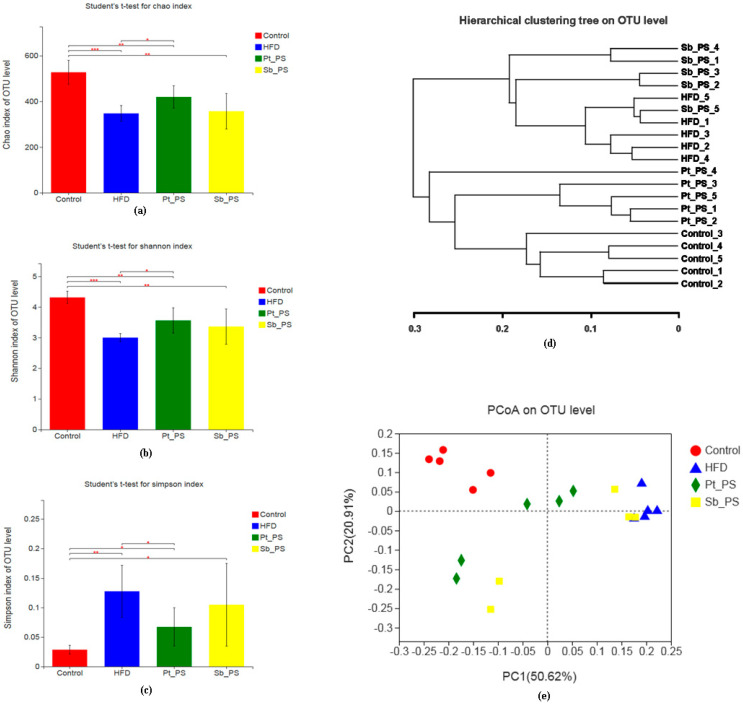
Effects of *Pt*-PS on the composition of the gut microbiota in HFD mice (*n* = 5). Univariate differential abundance of operational taxonomic units (OTUs) at the phylum level was tested by combining Fisher’s exact test and false discovery rate (FDR) to compare control, HFD, Sb-PS, and *Pt*-PS groups and mouse genotypes; *p* values were corrected with the Benjamini–Hochberg method to correct for the false discovery rate across multiple comparisons, which were generated using Metastats and considered significant for *p * <  0.05. The difference between control and HFD mice, HFD and *Pt*-egg oil groups was analyzed using Student’s test. (**a**) Chao index of OTU level; (**b**) Shannon index of OTU level; (**c**) Simpson index of OTU level; (**d**) hierarchical clustering tree at the OTU level; (**e**) Principal Coordinates Analysis (PCoA) at the OTU level. * *p* < 0.05, ** *p* < 0.01 *** *p* < 0.001 vs. HFD.

**Figure 4 marinedrugs-18-00483-f004:**
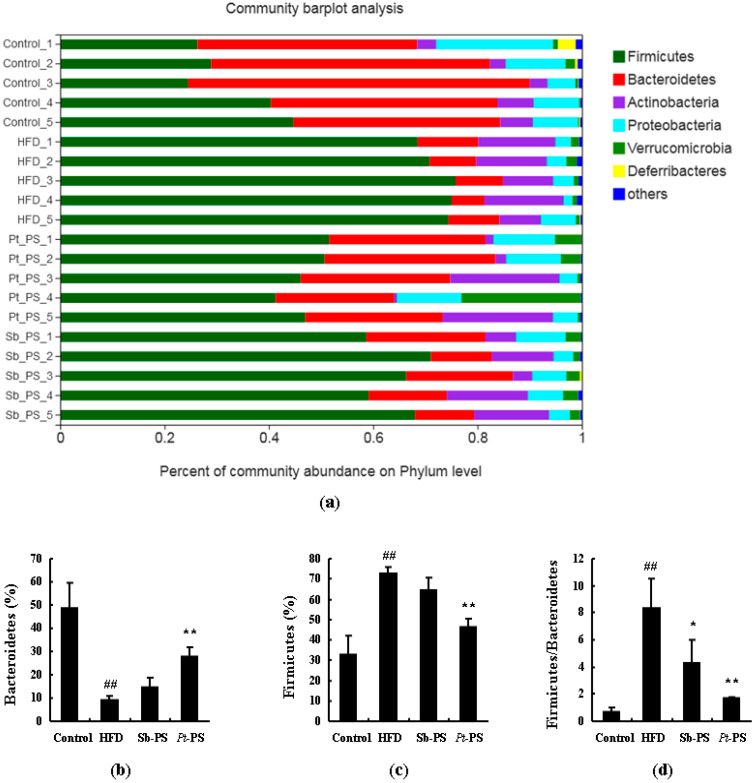
Effects of *Pt*-PS on the gut microbiota at the phylum level (*n* = 5). (**a**) Percent of community abundance at the phylum level; (**b**) Bacteroidetes at the phylum level in different groups; (**c**) Firmicutes at the phylum level in different groups; (**d**) the ratio of Firmicutes to Bacteroidetes t the phylum level in different groups. ^##^
*p* < 0.01 vs. control; * *p* < 0.05, ** *p* < 0.01 vs. HFD.

**Figure 5 marinedrugs-18-00483-f005:**
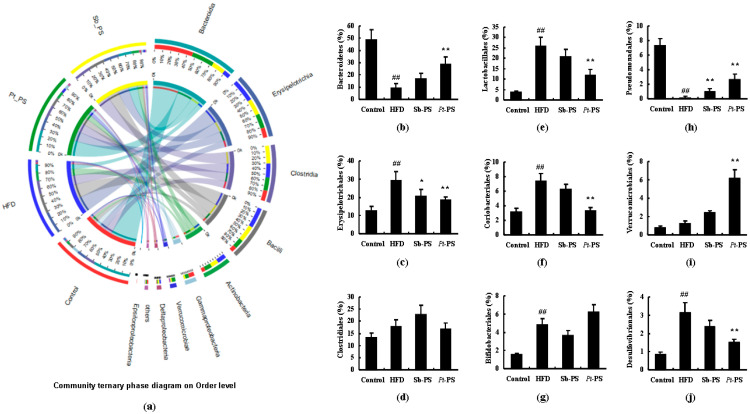
Effects of *Pt*-PS on the gut microbiota at the order level using a community ternary phase diagram (*n* = 5). (**a**) Percent of community abundance at the order level; (**b**) Bacteroidetes at the order level in different groups; (**c**) Drysipelotrichales at the order level in different groups; (**d**) Clostridiales at the order level in different groups; (**e**) Lactobacillales at the order level in different groups; (**f**) Coriobacteriales at the order level in different groups; (**g**) Bifidobacteriales at the order level in different groups; (**h**) Pseudomonadales at the order level in different groups; (**i**) Verrucomicrobiales at the order level in different groups; (**j**) Desulfovibrionales t the order level in different groups. ^##^
*p* < 0.01 vs. control; * *p* < 0.05, ** *p* < 0.01 vs. HFD.

**Figure 6 marinedrugs-18-00483-f006:**
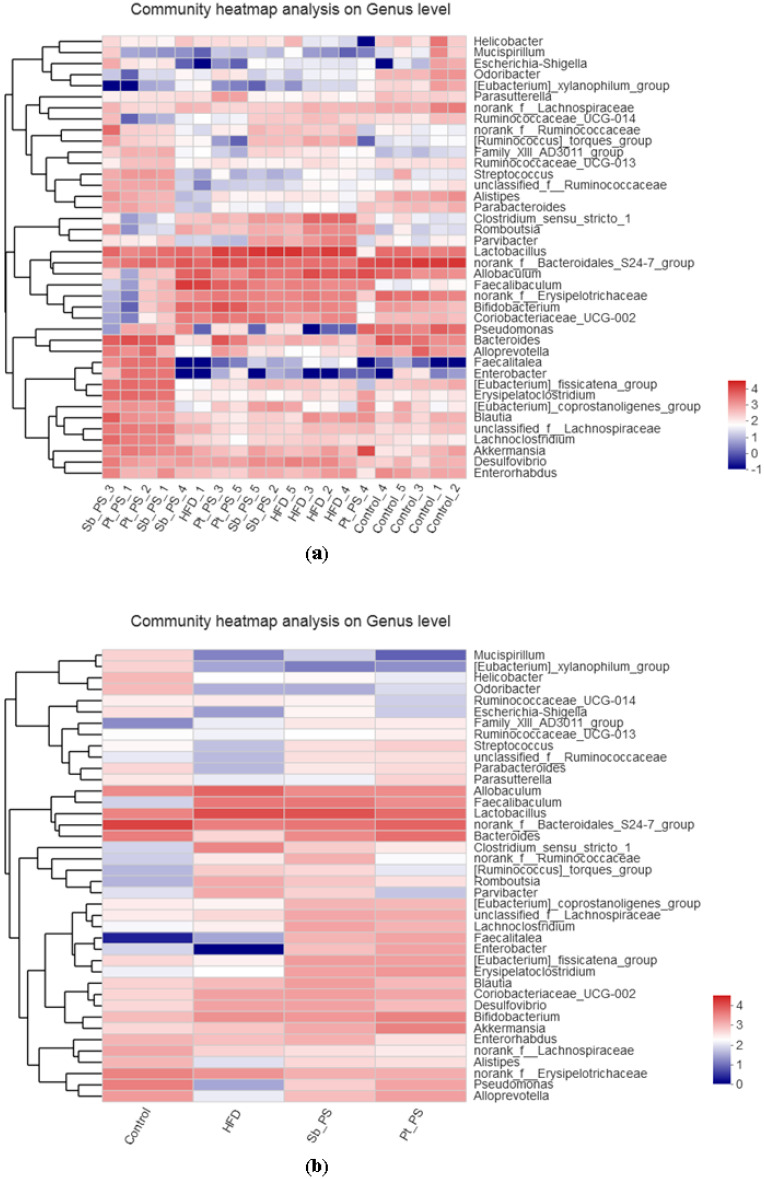
Response of the gut microbiota at the genus level to *Pt*-PS treatment (*n* = 5). Heatmap indicating the relative contributions of the top 40 dominant genera in each sample (**a**) and different groups (**b**). The heatmap is color-coded based on row Z-scores.

**Figure 7 marinedrugs-18-00483-f007:**
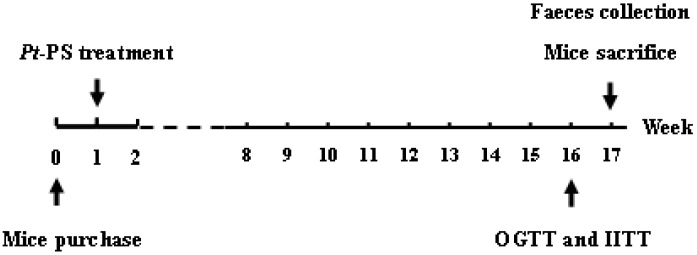
Animal experimental procedure.

**Table 1 marinedrugs-18-00483-t001:** Composition of fatty acids of *Pt*-PS.

Fatty Acids	Proportion (%)	Fatty Acids	Proportion (%)
C15:0	0.59 ± 0.08	C20:3	2.49 ± 0.27
C16:0	10.95 ± 1.57	C20:4	1.60 ± 0.49
C16:1	0.83 ± 0.01	C20:5	18.70 ± 3.32
C17:0	5.18 ± 0.84	C22:2	2.70 ± 0.56
C17:1	7.32 ± 0.86	C22:6	30.43 ± 2.08
C17:3	0.12 ± 0.01	C23:1	0.60 ± 0.11
C18:0	3.45 ± 0.61	Others	8.52 ± 1.40
C18:1	1.87 ± 0.30	∑SFA	22.19 ± 3.15
C18:2	0.59 ± 0.01	∑UFA	69.29 ± 5.20
C18:3	0.50 ± 0.02	∑MUFA	12.16 ± 2.24
C20:0	2.01 ± 0.54	∑PUFA	57.13 ± 5.63
C20:1	1.56 ± 0.39	∑(EPA + DHA)	50.13 ± 2.81

Data are presented as means ± SD (*n* = 6). Fatty acids were analyzed by gas chromatography. MUFA, monounsaturated fatty acids; *Pt*-PS, phosphatidylserine from *P. trituberculatus* eggs; PUFA, polyunsaturated fatty acids; SFA, saturated fatty acids; UFA, unsaturated fatty acids; EPA, eicosapentaenoic acid; DHA, docosahexaenoic acid.

**Table 2 marinedrugs-18-00483-t002:** Effect of *Pt*-PS on body weight, blood glucose, insulin, and adipokines in high-fat-diet (HFD)-fed mice.

	Control	HFD	Sb-PS	Pt-PS
Body weight gain (g)	12.21 ± 1.33	25.07 ± 2.67 ^##^	20.52 ± 1.86 *	16.61 ± 1.44 **
Food intake (g/w)	30.86 ± 2.75	25.95 ± 2.04 ^#^	25.67 ± 2.23	26.20 ± 2.49
Calorie intake (kcal/w)	126.38 ± 9.94	170.27 ± 11.36 ^#^	168.78 ± 12.47	171.44 ± 16.35
Perirenal fat weight (g)	0.20 ± 0.01	0.64 ± 0.07 ^##^	0.41 ± 0.03 **	0.31 ± 0.02 **
Epididymal fat weight (g)	0.58 ± 0.06	2.28 ± 0.30 ^##^	1.84 ± 0.21 *	1.17 ± 0.16 **
Abdominal subcutaneous fat weight (g)	0.19 ± 0.01	0.92 ± 0.09 ^##^	0.65 ± 0.07 *	0.45 ± 0.08 **
Hepatic weight (g)	1.13 ± 0.12	1.95 ± 0.14 ^##^	1.50 ± 0.20 *	1.36 ± 0.15 **
Fasting blood glucose (mmol/L)	7.73 ± 0.57	12.01 ± 0.94 ^##^	9.30 ± 0.87 *	8.48 ± 0.90 **
Serum insulin (mU/L)	9.26 ± 0.85	14.04 ± 1.32^##^	11.36 ± 0.89 *	10.42 ± 1.01**
HOMA-IR	3.20 ± 0.36	7.59 ± 0.84 ^##^	4.63 ± 0.52 **	3.96 ± 0.27 **
QUICKI	0.54 ± 0.02	0.45 ± 0.02 ^##^	0.49 ± 0.03	0.51 ± 0.01 *
Serum TNF-α (pg/mL)	63.58 ± 38.92	126.47 ± 47.61 ^##^	101.15 ± 40.03	86.34 ± 31.48 *
Serum adiponectin (ng/mL)	173.45 ± 15.21	82.67 ± 6.89 ^##^	144.73 ± 11.40 **	162.06 ± 9.74 **
Serum leptin (ng/mL)	0.18 ± 0.02	0.46 ± 0.04 ^##^	0.39 ± 0.05	0.24 ± 0.03 **
Serum resistin (ng/mL)	4.27 ± 0.30	11.26 ± 0.87 ^##^	7.85 ± 0.69 **	5.94 ± 0.82 **

Data are presented as mean ± S.D. Multiple comparisons were done using one-way ANOVA. The values of the first seven parameters were acquired from all animals (*n* = 14), while other parameters were determined from mice injected with normal saline (*n* = 7). ^#^
*p* < 0.05, ^##^
*p* < 0.01 vs. control; * *p* < 0.05, ** *p* < 0.01 vs. HFD group. Sb-PS, PS from soybean; HOMA-IR; QUICKI, quantitative insulin sensitivity check index.

**Table 3 marinedrugs-18-00483-t003:** Effect of *Pt*-PS on fecal lipopolysaccharide (LPS) and short-chain fatty acids (SCFAs) in insulin-resistant mice.

Parameters	Control	HFD	Sb-PS	Pt-PS
Fecal LPS (µg/g feces)	5.23 ± 0.64	13.47 ± 1.09 ^##^	11.67 ± 1.10	7.53 ± 0.77 **
Fecal acetate (mmol/L)	19.21 ± 2.84	8.80 ± 0.85 ^##^	11.36 ± 1.04 *	14.05 ± 1.37 **
Fecal propionate (mmol/L)	6.62 ± 0.60	3.27 ± 0.39 ^##^	3.76 ± 0.44	4.68 ± 0.41 *
Fecal butyrate (mmol/L)	1.82 ± 0.13	0.67 ± 0.08 ^##^	0.89 ± 0.10	1.43 ± 0.12 **

Data are presented as mean ± S.D. Multiple comparisons were done using one-way ANOVA. The mice were injected with normal saline before sacrifice (*n* = 7). ^##^
*p* < 0.01 vs. control; * *p* < 0.05, ** *p* < 0.01 vs. the HFD group.

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
