# Peer review of "Phosphatidylserine from Portunus trituberculatus Eggs Alleviates Insulin Resistance and Alters the Gut Microbiota in High-Fat-Diet-Fed Mice"

_marinedrugs, 2020, doi:10.3390/md18090483_

Round 1
Reviewer 1 Report
I appreciate the opportunity to review this manuscript on the effects of phosphatidylserine (PS) from Portunus trituberculatus (Pt) eggs on insulin resistance and gut microbiota in high-fat diet (HFD)-fed mice.
The genesis of this paper is the previous observation that egg oil isolated from Portunus trituberculatus improved hyperglycemia and hyperinsulinemia in HFD-fed mice and showed anti-obese effects through regulating gut microbiota.
In their paper, the authors have tested the hypothesis that phosphatidylserine from Pt eggs (has positive effects on insulin resistance and gut microbiota. Four groups were used: the control group (fed with a low-fat diet), HFD group, Sb-PS group (HFD and soybean-PS and Pt-PS group (HFD and Pt-PS).
Authors observed that Pt-PS counteracted the adverse effects of HFD on insulin resistance, the release of pro-inflammatory adipokines and dysbiosis.
I commend the authors for several strengths of their work, including addressing an interesting and timely question, careful planning and design of experiments, the proper conduct of experiments, and well-performed analysis.
The subject is in the range of the journal, and the manuscript is of clinical relevance. It is well written, and data are appropriately presented.
Considering these strengths, though, as I read the manuscript, I found some areas in which I would have appreciated greater clarity.
- There is insufficient information about the conducted research. For example, it is not mentioned that also soybean-PS was tested.
- Why was a low-fat diet used as a control rather than the standard diet?
- What was the reason to study soybean-PS effects? If the authors consider the studying soybean-PS necessary, they should refer to the difference in between Sb-PS and Sb-PS effects in the discussion.
- A flow chart showing the experimental procedure should be included as it makes the experimental setting more visible to the reader.
Author Response
Monday, 31 August 2020
Managing Editor
Marine Drugs
Dear reviewer,
Thank you for your effort on our manuscript, namely “Phosphatidylserine from Portunus trituberculatus eggs alleviates insulin resistance and alters gut microbiota in high fat diet-fed mice”. We also appreciate the meticulous review provided by your expert. Our responses to the comments are as follows.
I appreciate the opportunity to review this manuscript on the effects of phosphatidylserine (PS) from Portunus trituberculatus (Pt) eggs on insulin resistance and gut microbiota in high-fat diet (HFD)-fed mice. The genesis of this paper is the previous observation that egg oil isolated from Portunus trituberculatus improved hyperglycemia and hyperinsulinemia in HFD-fed mice and showed anti-obese effects through regulating gut microbiota. In their paper, the authors have tested the hypothesis that phosphatidylserine from Pt eggs (has positive effects on insulin resistance and gut microbiota. Four groups were used: the control group (fed with a low-fat diet), HFD group, Sb-PS group (HFD and soybean-PS and Pt-PS group (HFD and Pt-PS).Authors observed that Pt-PS counteracted the adverse effects of HFD on insulin resistance, the release of pro-inflammatory adipokines and dysbiosis.I commend the authors for several strengths of their work, including addressing an interesting and timely question, careful planning and design of experiments, the proper conduct of experiments, and well-performed analysis.The subject is in the range of the journal, and the manuscript is of clinical relevance. It is well written, and data are appropriately presented.
Considering these strengths, though, as I read the manuscript, I found some areas in which I would have appreciated greater clarity.
- There is insufficient information about the conducted research. For example, it is not mentioned that also soybean-PS was tested.
Answer: Thank you very much for the advice. Sb-PS was purchased from Shanghal Aladdin Biochemical Technology (S832149, Shanghai, China). Its fatty acids were clear, and therefore, the fatty acids were not tested.
- Why was a low-fat diet used as a control rather than the standard diet?
Answer: Thank you for the advice. In the experiment, we fed the control mice with the standard diet, while a mistake was in the manuscript. We have revised it as standard diet.
- What was the reason to study soybean-PS effects? If the authors consider the studying soybean-PS necessary, they should refer to the difference in between Pt-PS and Sb-PS effects in the discussion.
Answer: Thank you for the suggestion. We used Sb-PS as a positive control group. We also discussed the different activities of Pt-PS and Sb-PS in the second paragraph of the discussion section.
- A flow chart showing the experimental procedure should be included as it makes the experimental setting more visible to the reader.
Answer: Thank you very much for the suggestion. We have added a chart to explain the experimental procedure.
Thank you for your consideration. I look forward to hearing from you.
Best wishes
Yours sincerely,
Shiwei Hu

Reviewer 2 Report
The Authors describe the beneficial effects of Phosphatidylserine from Portunus trituberculatus (Pt-PS) (swimming crab) eggs on body weight, insulin resistance and other hematochemical and biochemical parameters of glycemic regulation in high fat diet exposed mice, compared to controls and to mice given soya bean phosphatidylserine. Moreover, they show the eubiotic effect of Pt-PS on mice gut microbiota, which is known to be altered in obese mice.
The study is very interesting and well designed and presented, also supported by figures and tables easily understandable and sufficiently easy to interpret. The Material and Methods section is very well presented and rich of data, though it's not so clear to understand the total number of subjects included (14 subdivided in 4 groups? 20, as shown in figure 4?).
The results show a remarkable efficacy of Pt-PS either on clinical or biochemical parameters of insulin resistance and provide novel means for efficacious prebiotic intervention which may restore eubiosis in obese subjects.
Discussion and Conclusion subheadings are well described and documented, as well as the list of references.
Author Response
Monday, 31 August 2020
Managing Editor
Marine Drugs
Dear reviewer,
Thank you for your effort on our manuscript, namely “Phosphatidylserine from Portunus trituberculatus eggs alleviates insulin resistance and alters gut microbiota in high fat diet-fed mice”. We also appreciate the meticulous review provided by your expert. Our responses to the comments are as follows.
The Authors describe the beneficial effects of Phosphatidylserine from Portunus trituberculatus (Pt-PS) (swimming crab) eggs on body weight, insulin resistance and other hematochemical and biochemical parameters of glycemic regulation in high fat diet exposed mice, compared to controls and to mice given soya bean phosphatidylserine. Moreover, they show the eubiotic effect of Pt-PS on mice gut microbiota, which is known to be altered in obese mice.
The study is very interesting and well designed and presented, also supported by figures and tables easily understandable and sufficiently easy to interpret. The Material and Methods section is very well presented and rich of data, though it's not so clear to understand the total number of subjects included (14 subdivided in 4 groups? 20, as shown in figure 4?).
Answer: Thank you very much for the suggestion. The total number of animal in each group was 14. The number of animals used to test OGTT and IITT was 7 in each group. Before sacrificed, 7 mice were suffered with normal saline stimulation in order to test total protein, while 4 mice or 3 mice suffered with different concentration of insulin stimulation in order to test phosphorylated protein and m-Glut4 protein. For gut microbiota analyze, we used 5 animals each group.
The results show a remarkable efficacy of Pt-PS either on clinical or biochemical parameters of insulin resistance and provide novel means for efficacious prebiotic intervention which may restore eubiosis in obese subjects.
Answer: Thank you very much for your appreciation and contribution to our manuscript.
Discussion and Conclusion subheadings are well described and documented, as well as the list of references.
Answer: Thank you very much for your appreciation and contribution to our manuscript.
Thank you for your consideration. I look forward to hearing from you.
Best wishes
Yours sincerely,
Shiwei Hu
